

# Effect of warm-up protocols using lower and higher loads on multiple-set back squat volume-load

Daniel Souza[1], Anderson Garcia Silva[1], Arthur Vale[2], Alana Pessoni[1], Luan Galvão[2], Murilo Augusto Araújo[2], Célio de Paula Júnior[3], Carlos Vieira[1], Amilton Vieira[4] and Paulo Gentil[1,2]

[1] Faculdade de Educação Física e Dança, Universidade Federal de Goiás, Goiania, Goias, Brazil
[2] Programa de Pós Graduação em Ciências da Saúde, Universidade Federal de Goiás, Goiania, Goias, Brazil
[3] Faculdade de Educação Física, Centro Universitário Uniaraguaia, Goiania, Goias, Brasil
[4] Faculdade de Educação Física, Universidade de Brasília, Brasilia, DF, Brazil

## ABSTRACT

**Background:** The present study aimed to investigate the effects of post-activation performance enhancement (PAPE) after three warm-up protocols on back squat performance in trained men.

**Methods:** Fourteen resistance-trained men performed conditioning activity (CA) with high-load (HL-CA), low-load (LL-CA), or usual specific warm-up as a control (CON). HL-CA consisted of one set of three repetitions with 90% of one repetition maximum (RM); LL-CA consisted of one set of six repetitions with 45% of 1 RM performed at maximal velocity; CON involved eight repetitions with 45% of 1 RM at controlled velocity. The participant's performance was measured using the total number of repetitions and volume load (reps × load × sets).

**Results:** There were no significant differences between warm-up for the total number of repetitions ($p = 0.17$) or total volume load ($p = 0.15$). There was no difference between conditions for the number of repetitions (main condition effect; $p = 0.17$); however, participants achieved a significantly higher volume load after HL-PAPE than after CON for the first set ($p = 0.04$).

**Conclusion:** High or low equated-load CA used as warm-up strategies did not potentiate subsequent performance enhancement in multiple-set back squat exercise performed until muscle failure in comparison with usual warm-up.

# INTRODUCTION

Resistance training (RT) adaptations depend on adequate manipulation and combining of several variables, such as repetitions, sets, and external loads (*Paoli & Bianco, 2012*; *Souza et al., 2018*). The product of repetitions, set, and load is described as volume load, which reflects the quantity of the external load lifted and is supposed

Corresponding author
Paulo Gentil,
paulogentil@hotmail.com

to be one of the most critical variables responsible for determining muscle adaptations (*Peterson et al., 2011*; *Nóbrega et al., 2023*). Based on this, coaches and athletes commonly seek strategies that increase volume load, as they could provide additional benefits to RT outcomes.

In this context, studies have investigated the effects of many strategies (*e.g.*, nutritional (*Spriet & Gibala, 2004*), mind-muscle connection (*Calatayud et al., 2016*), hypnosis (*Morton, 2003*), and advanced technique (*Krzysztofik et al., 2019*)) on performance and, consequently, on volume load. However, some strategies present controversial results, while others are costly and may be inaccessible to most RT practitioners (*Choi, Kim & Bae, 2022*). In this sense, warm-up protocols that promote post-activation performance enhancement (PAPE) have emerged as an inexpensive and feasible option for improving RT performance (*Boullosa, 2021*).

PAPE is evoked in response to a conditioning activity (CA), high-intensity muscle contractions performed before the main task (*Cuenca-Fernández et al., 2017*). The PAPE effect was initially associated with the post-activation potentiation (PAP) phenomenon, characterized by increased muscle contractile capacity evoked by an electrical stimulation resulting from a previous vigorous voluntary contraction (*Hodgson, Docherty & Robbins, 2005*; *Tillin & Bishop, 2009*). However, considering the divergent time course observed between PAP and PAPE after a CA, other physiological mechanisms might be considered to understand PAPE response, including increased muscle temperature, water content, and activation (*Blazevich & Babault, 2019*; *Fischer & Paternoster, 2024*).

There is evidence of PAPE in conventional RT involving lower (*Conrado de Freitas et al., 2021*) and upper (*Alves et al., 2021*; *Garbisu-Hualde, Gutierrez & Santos-Concejero, 2023*) body exercises after CA using high-load (HL-CA). These findings suggest that performing a few repetitions with heavier loads (*e.g.*, 80% 1 RM) during warm-up might improve acute exercise performance measured by increasing the number of repetitions performed in an RT session. However, warm-up involving loads closer to one repetition maximum (RM) may not be practical or safe for recreational RT practitioners, especially considering the high mechanical stress and need for supervision and specialized equipment (*Weakley et al., 2023*).

Previous studies have investigated the effects of PAPE using lower load CA (LL-CA) protocols on plyometric activities, such as jump squats (*Moir et al., 2011*) and sprints (*Rahimi, 2014*). Considering the possible mechanisms involved in PAPE (*Hodgson, Docherty & Robbins, 2005*; *Blazevich & Babault, 2019*), performing LL-CA using high-velocity contraction could induce greater muscle activation and increased temperature, contributing to PAPE similar to higher loads. Based on this, LL-CA might be a more practical and safer approach to overcome the barriers associated with HL-CA. However, there is no data on LL-CA effects on conventional RT performance. Therefore, this study aimed to compare the effects of CA with lower and higher loads on the number of repetitions performed in multiple-set back squat exercise. We hypothesized that HL-AC and LL-AC would be not different to improving performance during RT since both load schemes had the same volume load.

**Table 1** Characteristics of participants.

| ($n = 14$) | Mean ± standard deviation |
|---|---|
| Age (year) | 28.0 ± 3.8 |
| Bodymass OU mass (kg) | 80.3 ± 13.5 |
| Height (m) | 1.7 ± 0.1 |
| Resistance training experience (year) | 8.9 ± 4.7 |
| Relative strength (kg·kg$^{-1}$) | 1.4 ± 0.2 |

## MATERIALS AND METHODS

### Experimental design

This randomized crossover trial involved 14 resistance-trained men who visited the laboratory four times. Participants were instructed to abstain from exercise or strenuous physical activity involving the lower body for 48 h before the visits. During the first visit, anthropometric measurements and the back squat one-repetition maximum (1 RM) load were obtained. From the second to the fourth visit, the participants were randomly assigned to perform one of three experimental conditions with a 7-day interval between each session: HL-CA, LL-CA, and a control condition (CON). All experiments were performed between 8:00 and 10:00 a.m. The participants were instructed to maintain their nutritional habits during the study period (Table 1).

### Participants

Volunteers were recruited through social media and personal invitations among college students and the university's RT facility attendees. To be included in the study, the volunteers had to meet the following criteria: (i) 18 to 40 years old men, (ii) had been practicing RT regularly for at least 1 year, and (iii) be able to perform parallel back squats with proper technique until momentary muscle failure. Volunteers were not allowed to participate if they had any clinical condition or medical problems the study protocol could aggravate. The sample size was calculated based on the total number of repetitions effect size (d = 0.46) of a previous study (Conrado de Freitas et al., 2021), considering a power of 80% and an alpha of 5% using G*Power version 3.1.9.2 (Institute for Experimental Psychology, Dusseldorf, Germany). According to the calculation, a sample size of nine participants was necessary to detect the between-condition effects. Participants were informed about the study protocol, its risks, and benefits. They signed an informed consent form following the Declaration of Helsinki. The University of Goias Ethics Committee approved the study (approval # 56907716.5.0000.5083). All participants had more than 1 year of regular RT and could lift at least the load equivalent to their body mass during a 1 RM parallel back squat. The participant's physical characteristics, RT experience, and relative strength are presented in Table 2.

### One repetition maximum test (1 RM)

One RM back squat was measured using a Smith machine, according to the National Strength & Conditioning Association (2016) (NSCA) guidelines. Before the test, the

**Table 2 Description of nutrient and total energy intake for each experimental condition.**

| (n = 13) | HL-PAPE | LL-PAPE | CON | p |
|---|---|---|---|---|
| Carbohydrate (g) | 17.6 ± 11 | 18.5 ± 11 | 16.1 ± 10 | 0.27 |
| Protein (g) | 17.7 ± 13 | 17.0 ± 12 | 17.9 ± 12 | 0.96 |
| Lipid (g) | 17.6 ± 12 | 18.5 ± 11 | 16.1 ± 10 | 0.69 |
| Total energy (kcal) | 356 ± 170 | 405 ± 136 | 368 ± 158 | 0.60 |

Note:
PAPE, post-activation performance enhancement; HL-PAPE, PAPE with high-load; LL-PAPE, PAPE with low-load; CON, control group, usual specific warm-up.

participants performed a specific warm-up of 12 repetitions with a weight that they could typically lift 20 times. The weights were increased until the participants could perform only one repetition. Participants had a maximum of five attempts to achieve a 1 RM load with 3–5 min of rest between attempts. Participants received verbal encouragement throughout the test, and the same investigators conducted all testing procedures on the same machine with identical participant/equipment positioning.

## Parallel back squats protocol

The participants performed three sets of parallel back squats with 75% of 1 RM until momentary muscle failure. Participants started upright with their hips and knees completely extended and the bar positioned on the trapezius par above the scapular spines. The descendant movement was performed continuously until the thigh was parallel with the floor, and then the participant returned to initial position. A metronome-controlled movement velocity of 2 s was performed for each concentric and eccentric phases. In addition, an elastic device was used on the Smith machine to control the squat depth. The rest of the sets were fixed for 90 s. At least two exercise professionals supervised all sessions, and the participants were verbally encouraged to exert maximum effort. The total number of repetitions performed was recorded for each set and the volume load (repetitions × load) expressed in kilograms was calculated OU determined for further analysis.

## Conditioning activity protocols

Initially, the participants performed eight repetitions with 45% of 1 RM as usual warm-up at a controlled movement of 2 s for each concentric and eccentric phases. After 2 min of resting in the sitting position, the participants were randomly assigned by a draw to perform CON, HL-CA, or LL-CA condition. HL-CA involved one set of three repetitions with 90% of 1 RM, while LL-CA involved one set of six repetitions with 45% of 1 RM. After the CA protocols, the participants rested for 10 min and then performed the back squat protocol. CA protocols were equated by volume (load × repetitions), and both were performed with maximum intended velocity during concentric while controlling the eccentric phase for ~2 s. The adopted rest time after CA was based on greater effects verified from the previous literature review (*Wilson et al., 2013*). CON condition did not involve CA strategy and the participants performed the parallel back squats protocol 2 min after the usual warm-up.

## Statistical analysis

Data normality was confirmed using the Shapiro-Wilk test. Two-way repeated measures analysis of variance (ANOVA) was used to compare the experimental conditions (condition × sets). One-way ANOVA with repeated measures was used to compare the total measures accumulated over the three sets between the experimental conditions. When necessary, *post-hoc* comparisons were performed using Bonferroni adjustment. The data sphericity hypothesis was checked using the Mauchly test and corrected using Greenhouse-Geisser estimates when necessary. The effect size calculated by partial eta-squared ($\eta_p^2$) and observed power were obtained by ANOVA. Data were analyzed using SPSS 20.0 for Windows (SPSS, Inc., Chicago, IL).

## RESULTS

There was no significant difference for total number of repetitions between experimental conditions (HL-CA = 21 ± 5 (Confidence Interval 95% [18–24]) reps, LL-CA = 19 ± 5 (CI 95% [17–22]) reps, CON = 19 ± 6 (CI 95% [15–22]) reps, $p = 0.17$, $\eta_p^2 = 0.13$, power = 0.36). The mean difference between conditions for total repetitions was 1.8, 1.2, and 0.6 repetition for (HL-CA *vs*. CON), (HL-CA *vs*. LL-CA), and (LL-CA *vs*. CON), respectively. No significant main effect was detected when the number of repetitions was compared between conditions for each set (main condition effect, $p = 0.17$), whereas a significant main effect was detected for sets ($p < 0.001$). The first set presented higher repetitions than the second and third sets, and the second set presented higher repetitions than the third set (Fig. 1).

There was no significant difference for the total volume load was similar between the experimental conditions (HL-PAPE CA = 1,826 ± 690 (CI 95% [1,427–2,223]) kg, LL-PAPE CA = 1,723 ± 718 (CI 95% [1,307–2,137]) kg, CON = 1,668 ± 750 (CI 95% [1,235–2,100]) kg, $p = 0.15$, $\eta_p^2 = 0.13$, power = 0.38). The mean difference between conditions for total volume load was 158, 103, and 55 kg for (HL-CA *vs*. CON), (HL-CA *vs*. LL-CA), and (LL-CA *vs*. CON), respectively. No significant main effect was detected when the volume load was compared between conditions for each set ($p = 0.1$). However, the *post-hoc* comparison detected a significantly higher volume load for HL-PAP compared with CON for the first set ($p = 0.04$), as presented in Fig. 1. A significant main effect was detected for sets ($p < 0.001$), with the first set presenting a higher volume load than the second and third sets, and the second set presented a higher volume load than the third set (Fig. 1).

## DISCUSSION

This study aimed to compare the effects of CA protocols involving lower and higher loads equated by volume load on parallel back squat performance in resistance-trained men. As our main findings indicate, CA protocols involving lower and higher loads did not provide additional performance enhancement to multiple-set back squat performance compared to a usual warm-up. These results are of practical importance because performing a usual warm-up may represent a more practical and time-saving strategy for recreational RT practitioners.

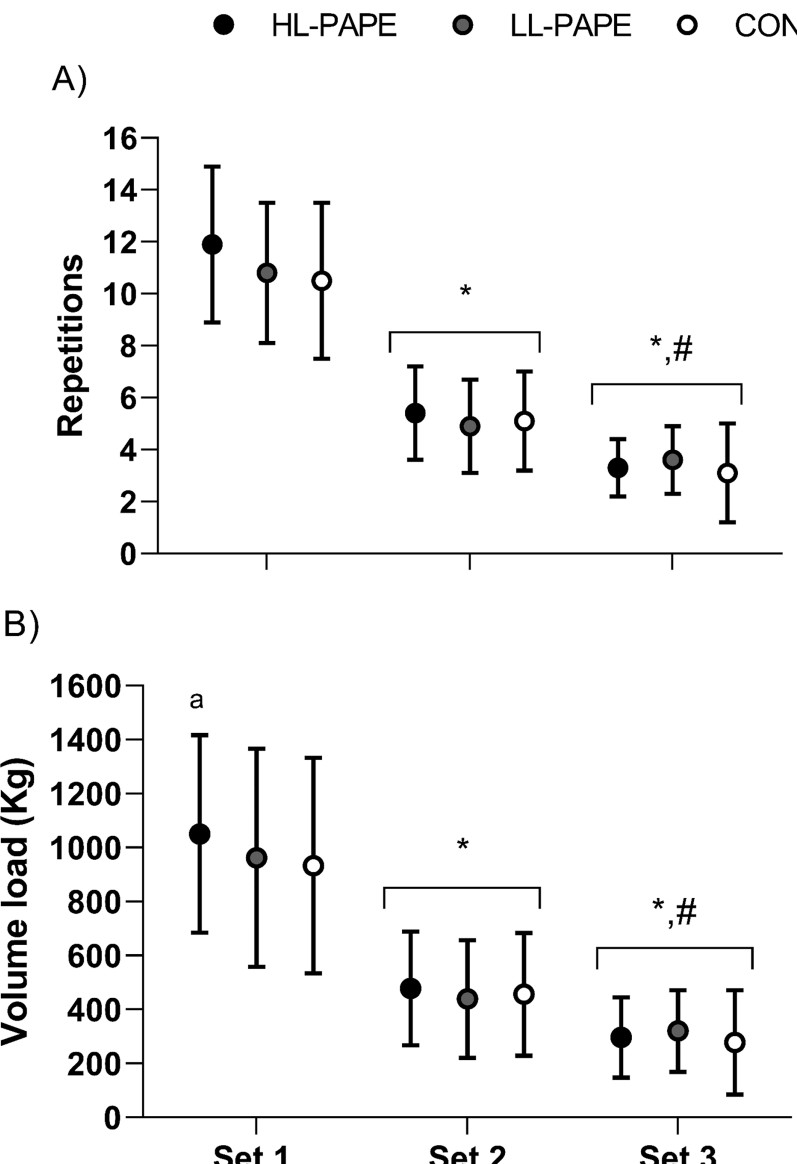

**Figure 1 Number of repetitions for each set between experimental conditions (A) and volume load for each set between experimental conditions (B).** HL-PAPE, High load post-activation performance enhancement; LL-PAPE, Low load post-activation performance enhancement; CON, Control. Values are presented as mean ± sd a Significant different compared with CON ($p < 0.05$); An asterisk (*) indicates significant difference compared with first set ($p < 0.05$); '#' indicated significant difference compared with second set ($p < 0.05$).

Previous studies have investigated the effects of CA involving resistance exercise during warm-up on subsequent physical performance enhancement, which usually include measures associated with muscle power and plyometric exercises such as sprints or a variety of jump tests (*Wilson et al., 2013*; *Chen et al., 2023*). However, few studies have investigated the potentiation effects of CA on specific RT session performance (*Boullosa, 2021*). In this context, a hallmarked characteristic of studies verifying PAPE on RT performance is the adoption of HL-CA (*Alves et al., 2021*;

*Conrado de Freitas et al., 2021*; *Garbisu-Hualde, Gutierrez & Santos-Concejero, 2023*), while studies investigating the effects of LL-CA are scarce. Thus, this is the first study to investigate the effect of CA using low-load on multiple-set back squat performed until muscle failure (*Moir et al., 2011*; *Rahimi, 2014*).

Considering that the purpose of CA is to engage muscle activation or potentiation, we performed LL-CA using higher-velocity movement to achieve higher motor unit recruitment in type II muscle fibers (*Oliveira & Negro, 2021*). However, although the performance verified after LL-CA was not different from the HL-CA, both CA protocols were also not statistically significant different from usual warm-up. On the other hand, when the VL is analyzed for each set separately, the higher VL verified for HL-CA in comparison to CON at the first set suggested the possibility that PAPE occurred only in a lower fatigue state, which has also been suggested by previous studies (*Alves et al., 2021*; *Conrado de Freitas et al., 2021*). Indeed, there was a trend of a higher number of repetitions for HL-CA compared to CON in the first set, which is in line with PAPE providing an extra aid to excel in performance in a specific 10-min window (*Wilson et al., 2013*). Therefore, although the PAPE effect was not evident over multiple sets, we cannot exclude their possible benefits in single-set tasks. On the other hand, considering that an ecological RT session aimed at muscle hypertrophy involves higher fatigue levels and a higher number of sets (*Morton, Colenso-Semple & Phillips, 2019*; *Souza, Barbalho & Gentil, 2020*). Adopting HL-CA could not represent a cost-effective strategy because the PAPE effect achieved in the first set can be lost over multiple sets.

Our results diverged from previous studies verifying the increased RT performance after HL-CA (*Alves et al., 2021*; *Conrado de Freitas et al., 2021*) specially in magnitude effect. When compared with CON condition, the present results from HL-CA represent approximately 9% in repetition performance improvement in back squats, whereas *Conrado de Freitas et al. (2021)* found improvement of 14%. Considering that usual warm-up could trigger possible mechanisms contributing to PAPE, such as changes in muscle temperature, cellular water content, and muscle activation (*Blazevich & Babault, 2019*), we cannot exclude the possibility that the usual warm-up might affect RT performance (*Bishop, 2003*; *Pearson & Hussain, 2014*) reducing the difference between HL-CA and CON condition. In addition, the adoption of lower velocity during the back squat protocol should reduce the PAPE response in our study since performance enhancement after CA is usually verified in tasks involving higher velocities, such as plyometrics (*McGowan et al., 2015*; *Blazevich & Babault, 2019*). It is important to point out that some evidence verified a lack of PAPE after conditioning activities in several performance context (*Brandenburg, 2005*; *Boullosa et al., 2018*; *Harrison et al., 2019*; *Jirovska et al., 2023*).

Another possible explanation for this divergence is that the PAPE verified in previous studies involving conventional RT performed for muscle failure may be biased by psychological factors and motivational state (*Armes et al., 2020*). The reduced performance observed over multiple sets in a previous study (*Conrado de Freitas et al., 2021*) for the CON group did not seem to align with multiple sets performed until momentary muscle failure (*Willardson & Burkett, 2006*). Considering the popularity of the benefits associated

with PAPE protocols in the sports context, volunteers that were aware of their participation in the experimental condition may be more likely to perform better, as previously suggested (*Lindberg et al., 2023*).

We also suggest that the divergence observed is associated with the different strength levels between the studied populations since PAPE seems more marked in stronger individuals (*Seitz & Haff, 2016*). However, the relative strength of our participants was similar to the previous study (*Conrado de Freitas et al., 2021*) $1.4 \pm 0.2$ (kg·kg$^{-1}$) *vs*. $1.3 \pm 0.25$ (kg·kg$^{-1}$), respectively. We complementary analyzed the correlation between relative strength and HL-CA/CON repetitions difference but did not find any association (rho $= -0.19$, $p = 0.5$).

The current study is original in comparing two loaded-equated warm-up protocols as a potential way to evoke PAPE response, and the inclusion of a usual warm-up condition serving as control. Moreover, the within-design helps to reduce the effects of many confounding variables, such as genetic and training background. However, the study is not free of limitations. Our findings should be interpreted cautiously because lower statistical power verified the between-conditions differences for the number of repetitions. While our study design considered the performance measuring in an ecological context, as recreational trainers usually perform multiple sets of dynamic exercises in RT, our study lacks other performance metrics like movement speed, force, or power. Although the tests procedures were largely carried out at the same time of the day, we might consider the possibility of daily fluctuation in 1 RM and performance influencing the results. Moreover, all participants in present study used the same interval transition after CA. Considering that time for optimal PAPE can vary between individuals (*McGowan et al., 2015*), determine each participant optimal transition duration to maximize the PAPE could change our results.

## CONCLUSION

Based on our results, using warm-up involving CA protocols with lower or higher loads did not potentiate subsequent performance enhancement over multiple sets of parallel back squat exercise performed until muscle failure in comparison with usual warm-up. Future studies should investigate other warm-up protocols, including performance metrics like movement speed, force, or power, as in-depth performance evaluation.

## ACKNOWLEDGEMENTS

We would like to thank all participants who volunteered their time to participate in the study.

### Funding

The authors received no funding for this work.

## Competing Interests

Daniel Souza, Anderson Silva, Arthur Vale, Alana Pessoni, Luan Galvão, Murilo Araujo, Celio de Paula Júnior, Carlos Vieira, Amilton Vieira and Paulo Gentil declare that they have no competing interests.

## Author Contributions

- Daniel Souza conceived and designed the experiments, performed the experiments, analyzed the data, prepared figures and/or tables, authored or reviewed drafts of the article, and approved the final draft.
- Anderson Garcia Silva conceived and designed the experiments, performed the experiments, authored or reviewed drafts of the article, and approved the final draft.
- Arthur Vale performed the experiments, authored or reviewed drafts of the article, and approved the final draft.
- Alana Pessoni performed the experiments, authored or reviewed drafts of the article, and approved the final draft.
- Luan Galvão performed the experiments, authored or reviewed drafts of the article, and approved the final draft.
- Murilo Augusto Araújo performed the experiments, authored or reviewed drafts of the article, and approved the final draft.
- Célio de Paula Júnior performed the experiments, authored or reviewed drafts of the article, and approved the final draft.
- Carlos Vieira analyzed the data, authored or reviewed drafts of the article, and approved the final draft.
- Amilton Vieira analyzed the data, authored or reviewed drafts of the article, and approved the final draft.
- Paulo Gentil conceived and designed the experiments, analyzed the data, prepared figures and/or tables, authored or reviewed drafts of the article, and approved the final draft.

## Human Ethics

The following information was supplied relating to ethical approvals (*i.e.*, approving body and any reference numbers):

The University of Goias Ethics Comittee grande approval to carry out the study (number: 56907716.5.0000.5083).

## Data Availability

The raw measurements are available in the Supplemental File.

## Supplemental Information

Supplemental information for this article can be found online at http://dx.doi.org/10.7717/peerj.17347#supplemental-information.

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
