# Peer review of "Effect of warm-up protocols using lower and higher loads on multiple-set back squat volume-load"

_PeerJ, doi:10.7717/peerj.17347_

## Round 0.1 · original submission · Major Revisions

Dear Authors,

Congratulations on the article.

Please carefully analyse the reviewers' comments and consider them to improve the manuscript.

Additionally, please respond to each of the comments point by point so that the reviewers, when analysing version 2 of the manuscript, compare the authors' responses with the reviewers' feedback.

Thank you.

Best regards.

**Language Note:** The review process has identified that the English language must be improved. PeerJ can provide language editing services - please contact us at [email protected] for pricing (be sure to provide your manuscript number and title). Alternatively, you should make your own arrangements to improve the language quality and provide details in your response letter. – PeerJ Staff

·

Basic reporting

Clear and unambiguous professional English is always used:
Meets a good level of English.

Sufficient bibliographic references, background/field context provided:
They are sufficient and adequate.

Structure of professional articles, figures, tables. Raw data shared:
Good structure, tables, and figures

Autonomous with results relevant to the hypotheses:
It does not comply, the results are poor, and with methodological questions: a single set per exercise is an issue that would be important to clarify why and then the way to measure performance is not an adequate way, it would have been important to measure the speed of the squat, or a vertical jump or a sprint, but the maximum number of repetitions does not have much evidence that indicates that it is a sensitive test

Experimental design

Original primary research within the Objectives and Scope of the journal:
I believe that it should be improved in order to be considered in the scope of the journal.

Well-defined, relevant, and meaningful research question. It states how research fills an identified knowledge gap:
The writing is adequate and shows the need to investigate the effect of PAPE with light loads, however, the form of performance measurement is not adequate.

Rigorous research conducted to a high technical and ethical standard:
It adequately complies with the ethical standard; however, the technician is poor.

Methods described in sufficient detail and information to replicate them:
Yes

Validity of the findings

Impact and novelty:
I believe that the results may lead to practical confusion due to poor methodological design.

All underlying data has been provided; They are robust, statistically robust, and controlled:
When talking about the research methodologically, procedures are carried out correctly, however, when talking about the methodology used in the intervention, the control is not very robust, the test is not adequate and therefore the finding is not very important.

Conclusions are well-expressed, linked to the original research question, and limited to supporting results:
Yes, however, because of the errors I mentioned above it could not be said that it is a true conclusion.

Additional comments

It's an interesting study, but other ways of measuring performance should be used, such as execution speed, vertical jump, or sprint.

Title: Good
Summary: good
Introduction: It is important to add information on why it is suspected that training with light loads at maximum velocity would be thought to be beneficial
Methods:
Line 108: Explicitly show the size of the effect used, since with the effect size of 0.50, which is the one I see in the study, I have a need for a larger sample.
Table 1: Present the demographic data in mean and standard deviation, did you make sure that these data had a normal distribution?
Table 2: You present the data in mean and standard deviation; did you make sure that these data had a normal distribution? And please indicate with which statistic you determined the p-value ¿ANOVA?
Lines 126 -134: How did they determine and control the depth of the squat?
Lines 140-144: Why was a single series decided?
Lines 147 -152: I don't understand the form of squat performance, the main effects of PAPE are in high-speed actions, and why do they set concentric velocity to two seconds? This is very slow.
Results:
I don't think that's what the test used to measure performance is designed for.

Reviewer 2 ·

Basic reporting

INTRODUCTION
-Line 62: This is not the original paper that presented the term & definition of the concept “PAPE”. This was attributed to a previous study conducted by “Cuenca-Fernández et al., 2017”.
-Line 66: What are those mechanisms? Physiological? Mechanical? Neurological?
-Lines 77-78: Why the myosin light-chain phosphorylation would impact PAPE effects when has been widely demonstrated that its effects are mostly dissipated after 30s? It may be possible that others mechanisms are responsible of modulating PAPE effects when applying both heavy/light loads? For instance, fast muscle fibers have been demonstrated to be highly dependent on muscle temperature due to its implication in reducing viscosity reduction, and these responses can be triggered either by sufficient exercise repetitions or by heavy resistance exercise. Authors should build (and include) their hypothesis in base of those facts.

Experimental design

METHODS
-Line 144: Why 10 min of rest between the conditioning exercise and the test? Why not 8-min or 12-min? Which previous literature authors based on to give this rest time? Why this rest time was not individualized?
-Line 212: Why this reference is in capital letters?

Validity of the findings

DISCUSSION
-Lines 229-238: Yes, I agree, authors should highlight this more. Even though Statistical analysis were not significant, PAPE strategies are mainly used to obtain an extra aid to excel in performance in competition on a specific window, but not during training. However, I can see “observable differences” or trends compared to control in Set-1 (which could be the one placed on the optimum window). As authors mentioned, the statistical power of this paper possibly compromised the assumptions reached, so I believe that authors should avoid creating strong arguments in opposite of observable trends that could become significant when increasing the sample or reducing the groups comparison.
-Line 222: Did the authors find any relation between the relative strength index and the responsiveness to specific warm-ups?
-Lines 246-247: Why authors included food intake data when nothing was discussed or performed with this information?
-Please add the meaning of the symbols included on the Figure.
-Why the Table 2 is necessary when only breakfast was monitored?

Additional comments

I strongly recommend authors to read these papers:

-Racinais, S., Cocking, S., & Périard, J. D. (2017). Sports and environmental temperature: from warming-up to heating-up. Temperature, 4(3), 227-257.
-Cuenca-Fernández, F., Smith, I. C., Jordan, M. J., MacIntosh, B. R., López-Contreras, G., Arellano, R., & Herzog, W. (2017). Nonlocalized postactivation performance enhancement (PAPE) effects in trained athletes: a pilot study. Applied Physiology, Nutrition, and Metabolism, 42(10), 1122-1125.

Reviewer 3 ·

Basic reporting

Please, see additional comment below.

Experimental design

Please, see additional comment below.

Validity of the findings

Please, see additional comment below.

Additional comments

Congratulations to the authors on the manuscript that aimed to analyze the post-activation performance enhancement (PAPE) after three warm-ups on back-squat repetitions. This study is interesting and tries to provide practical implications for researchers and sports professionals. This is the strength of the study, as well as the participants (number and level). However, I have some concerns regarding the novelty of the study (i.e., what this adds to the literature since there have been a lot of studies on this topic in the last few years) and regarding the procedures and outcomes. It is known that 1RM load varies daily and this is caused by a variation in performance. It is not clear how the authors dealt with this BIA issue and how it is guaranteed that the results were not compromised by this. Another concern is about the repetitions until the failure procedure. This has been debated in recent years and most of the recommendations are consensual on the non-need to use this method. However, for some purposes, this can be used. But the authors should better explain this, as well as how knowing that one specific warm-up caused to increase in the number of repetitions until failure can be useful for other resistance training programs, for example. Some specific comments are provided below.

- Please, revise the English writing and correct some errors throughout the manuscript (grammar and syntax).
- The authors found that there were no differences between conditions regarding the number of repetitions. Considering that each participant performed the three conditions with the same external load, should it be expected that the volume load would be different? Is not this a redundant analysis? Please, clarify your choices.
- Please, analyse each term used in the introduction according to their definition, removing or replacing them with other more appropriate ones. For example, load and volume load (load is the stimulus that is provided to the participant and volume is a part of the load), total load lifted, work (work is a measure of energy transfer caused by external force and measured in joule).
- In my opinion, the rationale of the introduction should be changed. The authors highlight the need for strategies to increase volume load and suggest PAPE to improve this. However, PAPE is a result of optimal warm-up strategies that should optimize performance. This performance optimization can be measured by some variables, such as volume, velocity, and so on. The authors choose to measure this by the number of repetitions until failure. Moreover, the authors should highlight the novelty of the study.
- Please, the authors should consider rewriting the purpose of the study. The authors reported that this study aimed to compare the effects of PAPE protocols. However, PAPE should be used to refer to the enhancement of measures of maximal strength, power, and speed following conditioning contractions. These are already effects caused by some exercise, in this case, different warm-ups. So, referring to the effects of PAPE is redundant (please, consider changing the title accordingly). Moreover, the variables of performance should be included in the purpose, and a hypothesis should be provided.
- Why did the authors choose to perform this warm-up in CON? Please, consider different warm-up strategies provided by the literature.
- Could the authors better explain the use of 10 min rest after warm-ups, as this can influence or compromise results?
- Can the authors better explain the use of 1min30s of rest between sets? Could this be insufficient to rest from repetitions until failure? Please, comment and discuss, if necessary.
- The data could benefit from additional analysis, such as confidence interval, differences, and effect sizes. Moreover, the authors included a “power” result, but it is not explained how this was determined. Please, clarify.
- Please, rewrite the “similar” word and meaning in the results. The non-existence of significant differences does not imply similarity. It only means that the difference was not proven.
- The discussion should be better supported by literature and a deeper analysis is needed. For example, most of the research in the last few years about the role of specific warm-ups for resistance training is not reported, appropriate limitations are not provided,

---

## Round 0.2 · Major Revisions

Dear author. The reviewers suggested changes/improvements to your work in order to go further in the process. Thank you. Best regards.

Reviewer 2 ·

Basic reporting

Thank you for conducting the changes.

My only concern was to identify which areas of the paper got changes and which ones were maintained with no changes. I've observed some comments of the rebuttal letter not applied to the paper or some changes that were not highlighted in red, so I don't know if all changes were indeed applied.

On the other hand, be aware that some information of the new paragraph of the discussion still contains typos like this "(ref)". Especially in this part, I cannot exactly identify how authors "tempered their discussion" about my comment regarding the window of opportunity in PAPE protocols.

Experimental design

No comment

Validity of the findings

No comment

Additional comments

No comment

Reviewer 3 ·

Basic reporting

The authors improved the manuscript according to suggestions. English has improved, literature references updated and article structure is correct. Nevertheless, some literature should be revised in the manuscript and references. Please, see my comments below.

Experimental design

The primary research is within the scope of the journal. The research purpose is now clear. Please, see my comments below.

Validity of the findings

Data results are now more supported but some clarifications are needed. Please, see my comments below.

Additional comments

Congratulations on the changes. In my opinion, the manuscript has improved a lot. Nevertheless, I feel that some minor issues should be taught by the authors, for example:
- I don't feel that the term "Schemes" is appropriate for the title and for the purpose...
- The hypothesis should be "different" or "not different" instead of equivalent
- What is a "controlled velocity" reported by the authors in methods, during PAPE? Please, clarify.
- The 2 seconds were for each or total concentric and eccentric?
- Please, be clear about the CON condition. This should be explained.
- The warm-up conditions were randomized? How?
- The symbol of partial eta squared is not correct. Please, change.
- Please, change the first paragraph of discussion (purpose) according to the new aim presented in the introduction.
- Please, revise literature citations throughout the text (format), and mainly in discussion. Please, also confirm that these are all in the references section.
- I think discussion should be improved, regarding syntax, reading flow. The ideas are not well sequenced and discussion should be easy to understand. Moreover, almost the same literature references are used to support findings and/or discuss.
- The authors should include the limitation of the study concerning day-to-day variation of 1RM and performance.
- In conclusion, the authors reported that "CA protocols did not increase the number of repetitions over multiple sets of parallel back squat exercise performed until muscle failure, regardless of light or heavy volume-equated load". This should be rewritten as well as every time this idea is presented. The "increase" imply the comparison with a previous state that was not assessed. Moreover, including the "increase" and the "regardless of light or heavy" causes some confusion. Please, consider to rewrite.

---

## Round 0.3 · Minor Revisions

Dear authors,

Please consider the feedback regarding minor issues from reviewer 3.

Thank you.

Best regards,

Reviewer 2 ·

Basic reporting

No comments

Experimental design

No comments

Validity of the findings

No comments

Additional comments

No comments

Reviewer 3 ·

Basic reporting

No comment

Experimental design

No comment.

Validity of the findings

No comment

Additional comments

The authors improved the manuscript according to previous suggestions. Nevertheless, I feel that some minor things should be clarified/changed and improved.
The conclusion presented by the authors reported that: "High or low equated-load warm-up strategies did not improve subsequent performance in multiple-set back squat exercise performed until muscle failure. " . The question is, did not improve in comparison to what? The warm-up do not intend to improve, but to maximize performance. Moreover, the comparison was to a control condition, where there is also a warm-up. So, the "not improved" is not clear, and the conclusion must be rephrased, please.
Moreover, considering that the study aim is to compare lower and higher loads on back squat performance, the authors could discuss more on this, comparing with recent findings on specific warm-up role and effects on resistance training/exercise. Nevertheless, all the manuscript has improved, and the authors made a great work.

---

## Round 0.4 · accepted · Accept

Dear Dr. Gentil,

Thank you for addressing the reviewers' comments.

Congratulations, the manuscript has been accepted for publication.


Reviewer 3 ·

Basic reporting

No comment

Experimental design

No comment

Validity of the findings

No comment

Additional comments

The authors addressed all my concerns. Congratulations on the work.